# Characterization of Microstructural Evolution in Heat-Affected Zone of Cu-Bearing Ultra-High-Strength Steel with Lamellar Microstructure

**DOI:** 10.3390/ma16020550

**Published:** 2023-01-05

**Authors:** Chao Fang, Chengning Li, Fengqin Ji, Wen Fu, Wenyi Hu, Xinjie Di

**Affiliations:** 1School of Materials Science and Engineering, Tianjin University, Tianjin 300350, China; 2Tianjin Key Laboratory of Advanced Joining Technology, Tianjin 300350, China; 3College of Aeronautical Engineering, Civil Aviation University of China, Tianjin 300300, China; 4State Key Laboratory of Solid Lubrication, Lanzhou Institute of Chemical Physics, Chinese Academy of Sciences, Lanzhou 730000, China

**Keywords:** ultra-high-strength steel, lamellar microstructure, Cu-rich precipitate, mechanical property, microstructure–property relation, welding, heat-affected zone

## Abstract

The advanced lamellar microstructure significantly improves the toughness of Cu-bearing ultra-high strength steel by delamination toughening (yield strength: 1370 MPa, impact toughness at −40 °C: 60 J). The lamellar microstructure affects the microstructure evolution of heat-affected zone (HAZ), resulting in separate distributions of lath martensite and granular bainite in the complete austenitizing zone and the formation of cluster fresh martensite in the partial austenitizing zone. The grain refinement and decrease in dislocation density are predominant features, especially for the complete austenitizing zone, where the grain is refined to 4.33 μm, and dislocation density is decreased by 27%. With the degree of austenitizing increase, the dissolution of Cu-rich precipitates (CRPs) aggravates during welding. A small fraction of CRPs in the complete austenitizing zone implies the onset of reprecipitation of CRPs. The reason for softening in HAZ is attributed to a combined effect of granular bainite forming, dislocation density decreasing, and CRPs dissolving. After PWTH, large numbered reprecipitation of coherent CRPs occurs, contributing to the hardness recovery of HAZ. Meanwhile, due to the high density of dislocation of lamellar microstructure inherited by partial austenitizing zone, coarsening of coherent CRPs is easy to occur, and various incoherent structures are observed.

## 1. Introduction

Advanced ultra-high strength (UHS) steels are greatly desirable to meet the demand for weight reduction and energy conservation in widespread engineering applications [1,2]. The traditional composition design is to add high carbon and alloy content, which leads to serious deterioration of toughness and weldability in UHS steel [1,3]. Cu-rich precipitates (CRPs) have been proven to be the most effective methods to provide high strength while keeping the low carbon content (<0.06 wt%) in steels [4,5,6,7]. Therefore, the contradiction between strength and weldability can be overcome. High density, small size, and coherent structure CRPs usually cause the peak strength by dislocation shearing mechanism [8,9,10]. Unfortunately, attaining high strength enhancement by coherent CRPs is usually accompanied by a sharp decrease in low-temperature toughness [11,12,13]. Recently, an advanced microstructure of lamellar microstructure has been proposed to achieve delamination toughening [14,15,16,17]. During impact, the interfacial delamination of lamellar interfaces effectively lowers the three-dimensional stress at the crack tip, resulting in crack blunted, improving the toughness. It is expected that a combination of ultra-high strength and suitable toughness can be achieved in Cu-rich precipitation to strengthen UHS steel with lamellar microstructure.

The heat-affected zone (HAZ) undergoes complex phase transformation leading to an inhomogeneous microstructure, which breaks the balance of strength and toughness. In traditional UHS steel with high carbon content, the formation of coarse grain size and brittle microstructure leads to an increase in hardness [1]. In Cu-bearing UHS steel, however, the insufficient thermal stability of CRPs is the main drawback [3]. Coarsening and dissolution of CRPs easily occur, leading to a significant loss of strength [18,19]. In addition, lamellar microstructure has a high density of dislocation. After welding, a significant reduction of dislocation density accompanied by phase transformation might aggravate the softening of HAZ. Therefore, the initial microstructure of base material might play an important role in the properties of HAZ. Fundamental precipitation and microstructure evolution behaviors are critical for the prediction and control of HAZ mechanical properties.

In this work, by applying low finish rolling temperature in a conventional hot rolling line, a lamellar grain structure is developed in Cu-rich precipitation-strengthened UHS steel. The evolution behaviors of the lamellar microstructure and precipitates during welding are investigated by means of scanning electron microscopy (SEM), electron back-scattered diffraction (EBSD), and transmission electron microscopy (TEM). The correlation between the microstructure, precipitates, and mechanical property evolution in the HAZ is elucidated. Furthermore, the effect of post-weld heat treatment on precipitates and properties is discussed in detail.

## 2. Materials and Methods

The experimental steel was melted in a 150 kg vacuum induction melting furnace and cast into an alloy ingot. The cast ingot was austenitized at 1200 °C for 2 h and then was hot rolled with 45% reduction at ~1150 °C and 78% reduction at ~750 °C to a final thickness of 12 mm by using a Φ450 mm two high rolling mill. After rolling, the plates were immediately cooled to 200 °C controlled by a water cooling spray cooling system with a cooling rate of ~50 °C/s, as shown in Figure 1. Based on our previous studies [6], aging at 550 °C for 2 h can provide a significant precipitation enhancement of strength. Thus, experimental steels were aged at 550 °C for 2 h, followed by air cooling to prepare the base material (BM). Table 1 lists the chemical composition obtained by spectroscopic analysis using DT-100 instruments. According to Graville et al. equation [20], the carbon equivalent of experimental steel is 0.295%, indicating suitable weldability. Figure 2 shows that the critical austenitization start temperature (A_C1_) and finish temperatures (A_C3_) of the experimental steel were estimated at 755 ± 10 °C and 880 ± 10 °C, respectively.

The specimens with a thickness of 10 mm were welded by applying the gas tungsten arc welding (GTAW) method in a high-purity argon atmosphere. Welding was conducted along the rolling direction with a heat input of ~21.6 kJ/cm. The current, voltage, welding speed, and gas flow are 228 A, 14.2 V, 1.5 mm/s, and 20 L/min, respectively. K-type direct contact thermocouples were used to measure the thermal cycle histories during welding. The as-welded specimens, labeled AW, were post-weld heat treated at 550 °C for 30 min, labeled PWHT.

The transverse cross-section of the weld joint was ground, polished, and etched with 3% Nital solution for ~5 s to reveal the microscopic characteristics using a JEOL JSM7800F SEM. To ensure the accuracy of analyzing the microstructure and precipitate features of different HAZ subregions, TEM samples were prepared by carefully cutting out 5 mm diameter discs with a specific HAZ subregion as the center. The discs were ground to 50 μm and then twin-jet electropolished within a solution of 90% acetic acid and 10% perchloric acid alcohol solution at −40 °C before observation under a Tecnai F30 TEM. EBSD specimens were electrochemically polished with 5% perchloric acid at 30 V at ambient temperature. EBSD orientation maps were constructed at a step size of 0.2 μm. The TSL OIM analysis software was employed for EBSD data analysis.

According to ASTM A370, tensile tests were conducted on specimens with Φ6.25 mm and a gauge length of 25 mm along the rolling direction at a crosshead speed of 0.25 mm/min at ambient temperature. According to the ASTM E23, the impact toughness was evaluated on standard Charpy V-notch impact specimens with 55 × 10 × 10 mm in size machined. The impact direction was parallel to the transverse direction. Hardness measurements were conducted on the polished surface of the samples using a Vickers hardness with a load of 1 kg for 15 s. Nanoindentation testing was carried out to investigate the microscale mechanical properties of the HAZ by using Hysitron TI Premier with a Berkovich diamond tip. To avoid the size effect, constant and high load values were selected. Load control testing mode is used with a maximum load of 10 mN, loading rate of 1 mN/s, holding time of 10 s, and unloading time of 1 mN/s. Indentation positions are traced with the aid of SEM and EBSD technology.

## 3. Results and Discussion

### 3.1. Microstructure Characteristics and Mechanical Properties of the Base Material

The microstructure of base material is a lamellar structure composed of deformed lath bainite (LB) (Figure 3a,b). The average grain size is ~7.54 μm. Figure 3c displays the gradient of the kernel average misorientation (KAM) value, which characterizes the local dislocation gradient. Figure 3d shows many dislocation tangles at the intersection of the bainite laths. The average size of bainite laths is ~290 nm. Selected area electron diffraction (SAED) along the [001]_α_ direction shows weak (100)-type superlattice reflections (Figure 3e), which are confirmed to be derived from B2 structure precipitates [12,21]. These B2 CRPs are coherent with the α-Fe matrix; therefore, they are difficult to directly observe in bright field micrography. The dark field (Figure 3c) morphology taken from the (010)_B2_ reflection exhibits nanoscale B2 CRPs precipitated in the experimental steel (Figure 3f).

Figure 4 shows the results of tensile and impact tests. The yield strength is calculated by stress at 0.02% strain. The yield strength of as-rolled steel reaches 932 ± 7 MPa. After aging treatment, the yield strength increases to 1370 ± 7 MPa. The effect of B2 CRPs as impediments of dislocation movements results in a significant enhancement of yield strength (~438 MPa). It is noted that a high elongation (~16.5%) is still obtained after aging treatment. This is because B2 coherent CRPs allow dislocations to shear through rather than acting as an impenetrable barrier to dislocation motion [21,22,23]. Due to delamination activated by the lamellar microstructure, low-temperature toughness at −40 °C is ~60 J, as presented in the insert of Figure 4b. In conclusion, a satisfactory combination of strength, ductility, and toughness is achieved in Cu-bearing UHS steel with a lamellar structure.

### 3.2. Microstructure Evolution of HAZ during Welding and PWHT

Figure 5 shows a schematic illustration of the welding process. According to the thermal cycling histories (Figure 5b), the HAZ can be divided into two subregions. The region with a peak temperature of 946 °C is a complete austenitizing zone, which is composed of granular bainite (GB) and lath martensite (LM) (Figure 6b). The other region with a peak temperature of 815 °C is a partial austenitizing zone, which consists of retained deformed LB and fresh martensite (FM) (Figure 6c). After PWHT, the microstructure of the complete austenitizing zone is still composed of GB and LB (Figure 6d), and the partial austenitizing zone is characterized by FM (Figure 6e), which suggests that PWHT has little effect on the microstructure.

Carbon is known as the most important element to affect the stability and hardenability of austenite. At low peak temperatures, reverted austenite grains are enriched in carbon and converted into FM in the partial austenitizing zone. With the increase in peak temperature, the volume fraction of reverted austenite increases, and elements diffuse more uniformly, resulting in the carbon content of reverted austenite decreasing [24,25]. Consequently, martensite transformation is suppressed while bainite transformation is promoted in the complete austenitizing zone. Moreover, the cooling rate in the complete austenitizing zone is also beneficial to form bainitic according to the CCT diagram (Figure 7). Therefore, the mixture microstructure of GB and LM is obtained in the complete austenitizing zone. Elongated prior austenite grain boundaries preferentially act as reverted austenite nucleation sites due to large free energy [24,26]. As a result, preferential austenite nucleation in the special region leads to separated distributions of the carbon-rich zone and carbon-poor zone, resulting in separated distributions of LM and GB. Similarly, FM shows a clustered distribution along grain boundaries.

Figure 8 shows the grain size distribution obtained from EBSD results. In the complete austenitizing zone, the large deformed LB grains (7.54 μm) are fully transformed into fine grains (~4.33 μm), presented in Figure 8a. In the partial austenitizing zone, the average grain size is ~4.57 μm (Figure 8b). There are nearly 50% of grains with a size smaller than 4 μm caused by the formation of fine FM. The deformed microstructure has a larger deformation and stores energy, which can provide more nucleation driving force. Consequently, faster and more nucleation of fresh grains results in grain refinement. In comparison, when the initial microstructure is without deformed, only a slight refinement can be obtained by investigating different austenite transformation degrees [25]. Grain refinement aroused by initial lamellar microstructure can be expected to improve the toughness of HAZ.

Figure 9 illustrates the KAM results of different HAZ subregions. The density of geometrically necessary dislocations can be estimated by the local average misorientation *θ*_KAM,_ which reflects the lattice curvature. The equation used to calculate GND densities *ρ*^GND^ is as follows:(1)$$\rho^{GND}=\frac{2\theta_{\mathrm{KAM}}}{\mu b}$$
where *θ*_KAM_ is the average misorientation angle across dislocation boundaries, *μ* is the average spacing of dislocation boundaries, and b is the magnitude of the Burgers vector. *θ*_KAM_ and *μ* are identified by the average value of KAM and the scanning step size, respectively. The average KAM value of the BM is 1.33, so its dislocation density is 5.32 × 10^12^ cm^−2^. After welding, the dislocation density of the complete austenitizing zone decreases to 3.87 × 10^12^ cm^−2^ (decreased by 27%) while that of the partial austenitizing zone reduces to 4.84 × 10^12^ cm^−2^ (decreased by 9%). There is more decrease in dislocation density as austenitizing degree increases. By comparison, for initial microstructure without deformation, the dislocation density presents an uptrend with the continuous increment of austenitizing degree [25]. Therefore, it can turn out that the initial high density of dislocation in lamellar microstructure is easily consumed during welding, which is difficult to be remedied by dislocation caused by martensitic transformation. This reduction of dislocation density is a significant problem for steel with a deformed lamellar microstructure, especially for the complete austenitizing zone.

### 3.3. Precipitates Evolution of HAZ during Welding and PWHT

Figure 10 shows the results of the TEM analysis of the AW joint. SAED patterns present (100)_B2_ superlattice diffraction spots of two HAZ subregions (Figure 10b,e), which indicates the presence of B2 CRPs. However, the intensity of superlattice diffraction spots in the complete austenitizing zone is very weak. The dark field micrography reveals only a small amount of B2 CRPs in the complete austenitizing zone (Figure 10c) but a relatively high density of B2 CRPs in the partial austenitizing zone (Figure 10f). Compared with BM (Figure 3f), the number density of CRPs decreases obviously. By simulation calculation, the critical dissolution temperature of CRPs is 744 °C (Figure 11). The peak temperature of two HAZ subregions is higher than 744 °C. Therefore, the dissolution of CRPs is inevitable. Owing to the greater solubility of Cu in austenite rather than ferrite, the dissolution of CRPs should be controlled by the austenitizing degree. CRPs could fully dissolve during reheating in the complete austenitizing zone, whereas partial dissolution of CRPs occurs in the partial austenitizing zone. The presence of a few fine CRPs in the complete austenitizing zone implies the occurrence of reprecipitation. Unlike carbide precipitates (such as NbC and TiC) which start to precipitate within the austenite region [27]. CRPs do not reprecipitate in austenite, avoiding the formation of coarse particles. Moreover, the rapid heating rate also does not provide a long holding time at the high-temperature stage. Consequently, coarse CRPs are not observed in HAZ.

After PWHT at 550 °C for 1 h, superlattice diffraction spots are also found (Figure 12b,e). The dark field micrographs present an increased number density of B2 CRPs in two HAZ subregions (Figure 12c,f), which proves that PWHT is an effective method to promote the reprecipitation of B2 CRPs. Furthermore, bright field observations present large CRPs with a size of ~10 nm and ~24 nm in the partial austenitizing zone (Figure 13 and Figure 14). Many studies have demonstrated that structure evolution is accompanied by coarsening of CRPs: B2 structure (less than 5 nm)→9R structure (4–17 nm)→3R structure (18–40 nm)→FCC structure (>40 nm) [10,28]. Accordingly, CRPs with a size of ~10 nm and ~24 nm might be 9R and 3R incoherent CRPs, respectively. Meanwhile, the EDS results reveal that the increase in Cu content is accompanied by the size increase in CRPs. Figure 15 shows the high-resolution morphology of CRP along the [111]_α_ direction. The CRP exhibits an elongated morphology with a long axis of approximately 17.72 nm. Many moire’ fringes in the CRP are parallel to the [1$\bar{1}$0]_α_ direction. These moire’ fringes are aroused by lattice overlap. Figure 15b and c show Fast Fourier Transforms patterns of the matrix and precipitates, respectively. The patterns and moire’ fringes are effective evidence for the formation of incoherent FCT structure [29].

It is concluded that suggest that coarsening of CRPs easily occurs in the partial austenitizing zone. Figure 9b shows that the high dislocation density of the initial lamellar microstructure is inherited in the partial austenitizing zone. Many studies demonstrated high density of dislocations can promote short-range diffusion of Cu atoms [29,30]. Therefore, for the steel with lamellar structure, the coarsen and incoherent structure evolution of CRPs are easier to occur during PWHT. However, in traditional steel without lamellar microstructure, the PWHT process (550 °C for 1 h) leads to the reprecipitation of CRPs with fine size and high number density, similar to that in the original microstructure [31]. In conclusion, the coarsen and incoherent structure evolution of CRPs should raise a concern. An appropriate PWHT process needs to be further studied to avoid the softening of the partial austenitizing zone after PWHT.

### 3.4. Effect of Microstructure on Hardness and Micromechanical Properties

Figure 16a shows the distribution of hardness in the AW joint. Compared with the BM (~440 HV_1_), a significant reduction in hardness is produced in the HAZ. Furthermore, the complete austenitizing zone shows a lower hardness (~320 HV_1_) than the partial austenitizing zone (~370 HV_1_). Compared with other UHS steel (for example, BA160 [18], NU140 [32], S960 [33], and S1100 [34]), the serious softening of HAZ is a unique feature of steel with lamellar structure, especially for the complete austenitizing zone (decreased by 28%). This is because of a combined effect of soft GB forming, dislocation density reduction, and CRP dissolution. After the PWHT process, Figure 16b shows that a homogeneous and improved hardness is obtained (~418 HV_1_), which is caused by large numbered reprecipitation of CRPs. However, there is still a loss of hardness in HAZ compared to that of BM.

Utilizing the EBSD technique, each indent of the nanoindentation tests is tracked. Micromechanical properties, including the nano-hardness (H) and elastic modulus (EM), are marked in Figure 17 and Figure 18. Figure 19 exhibits the corresponding load-displacement curves of different indents. The shorter displacement means the higher nano-hardness. In the AW joint, the partial austenitizing zone shows higher average values of micromechanical properties (H: ~2.45 GPa, EM: ~216.04) than that of the complete austenitizing zone (H: ~3.67 GPa, EM: 236.61). The difference in nano-hardness is consistent with the conclusion of Vickers hardness testing (Figure 16a). In addition, the nanoindentation test curves of the complete austenitizing zone exhibit a high degree of dispersion (Figure 18), indicating the inhomogeneous mechanical properties of the microstructure for the complete austenitizing zone. For the PWHT joint, the nano-hardness and elastic modulus are improved significantly. The average value of the nano-hardness in the complete austenitizing zone and partial austenitizing zone reaches ~4.15 GPa and ~4.96 GPa, respectively. Notably, the dispersion of the nanoindentation test curves decreases (Figure 19), which implies that the performance uniformity of the complete austenitizing zone is improved. The reasons for micromechanical properties changes are explained in detail as follows.

Nano-hardness is related to the microstructure type, grain orientation, and grain boundaries. First, there is a great difference in nano-hardness between various microstructures. In the complete austenitizing zone, LM (indent 1) shows higher nano-hardness than GB (indent 4), as shown in Figure 17a. The hardness of FM (indent 1) in the partial austenitizing zone is larger than that of retained deformed LB (indents 2, 3, and 4) marked in Figure 17b. Second, grain orientation also has an important effect on nano-hardness. For example, indents 2 and 3 are both located on the original LB microstructure (Figure 17b), but there is a large difference in the hardness and elastic modulus. Generally, bluish violet (close to the <111> orientation) grains exhibit higher hardness and modulus than other oriented grains. Third, the presence of grain boundaries could be a major reason for the larger hardness. Indent 2 is located on multiple grain boundaries (Figure 17a) and shows an extremely high nano-hardness (~3.58 GPa) than other indents, which could be ascribed to the grain boundaries providing deformation resistance, resulting in their high hardness [32].

The higher nano-hardness is not always accompanied by higher elastic modulus, such as indents 1 and 3 of Figure 17a and b. This is because elastic modulus is a parameter that is not sensitive to microstructure type and only reflects atomic bonding strength. However, a large elastic modulus is obtained when the indent is on HAGBs. Grain boundaries usually have a high content of C, Cu, Mn, Ni, and Al in Cu-bearing steel [35]. These elements have smaller atomic diameters than Fe atoms, leading to the atomic spacing decreasing. Consequently, the atoms become more tightly bound, resulting in elastic modulus increases. Figure 18 shows that an enhancement of the elastic modulus (~11) is observed in the complete austenitizing zone. Previous research has proposed that CRPs have a higher elastic modulus than the matrix, inducing modulus strengthening [5]. Large numbered reprecipitation of CRPs can be the major reason for the increase in elastic modulus.

## 4. Conclusions

1. The microstructure of Cu-bearing UHS steel consists of deformed lath bainite that shows elongated lamellar grain structure by applying low finish rolling temperatures at 750 °C. After aging, the yield strength and elongation reach ~1370 MPa and ~16.7%, respectively. A high low-temperature toughness (60 J at −40 °C) is induced by delamination toughening;

2. The advanced lamellar microstructure significantly influences the transformation behaviors during welding. A mixture of GB and LM in the complete austenitizing zone and FM in the partial austenitizing zone is observed. The lamellar microstructure gives rise to significant refinement, especially for the complete austenitizing zone (~4.33 μm). Meanwhile, the presence of elongated grain boundaries influences the distribution of microstructures, leading to the formation of separated GB zones in the complete austenitizing zone and a cluster distribution of FM in the partial austenitizing zone. Moreover, the initial high density of dislocation is decreased, especially for the complete austenitizing zone (decreased by 27%). The more reduction of dislocation density with the increase in austenitizing degree;

3. CRPs are shown to dissolve during welding, resulting in the reduction of number density. With the increase in austenite transformation degrees, the dissolution of CRPs aggravates. The coarsening of CRPs does not occur due to rapid heating and cooling rate. After PWHT, CRPs with B2 structure reprecipitate in HAZ. Furthermore, for the partial austenitizing zone, the coarsening of CRPs promoted by a high density of dislocation is observed. B2-coherent CRPs with a size of less than 5 nm are converted into an incoherent structure, including 9R (size of ~10 nm), 3R (size of ~24 nm), and FCT (size of ~18 nm) CRPs;

4. The softening of HAZ is a predominant problem after welding, especially for the complete austenitizing zone, where hardness is decreased by 28%. This is because of the combined effects of the formation of granular bainite, the decrease in dislocation density, and the dissolution of Cu-rich precipitates. The PWHT has been approved to be an effective method for recovery of hardness (~420 HV_1_) and enhancement of elastic modulus (~11), which can be ascribed mainly to large numbered reprecipitation of CRPs. However, due to the disappearance of lamellar microstructure, the hardness of HAZ is also less than that of BM.

## 5. Recommendations

Due to the coexistence of reprecipitation and coarsening of CRPs, the PWHT process needs to be designed carefully to avoid the inhomogeneous mechanical properties of various HAZ subregions. On the other hand, the transformation of lamellar microstructure should lead to the disappearance of delamination toughening in HAZ. When recovering strength after PWHT, the toughness changes of HAZ need to be further studied.

## Figures and Tables

**Figure 1 materials-16-00550-f001:**
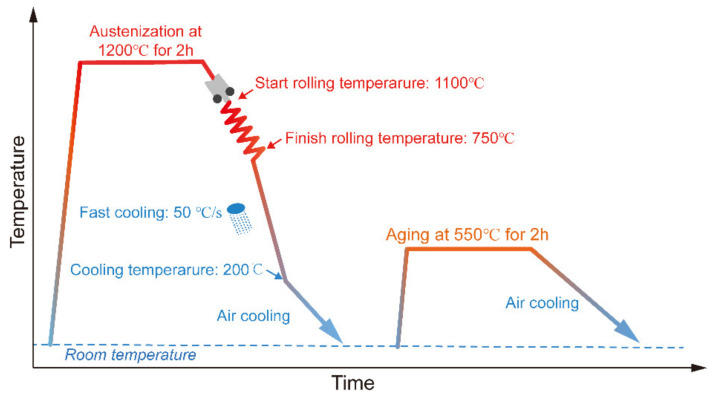
Schematic illustration of the TMCP and aging process for the experimental steel.

**Figure 2 materials-16-00550-f002:**
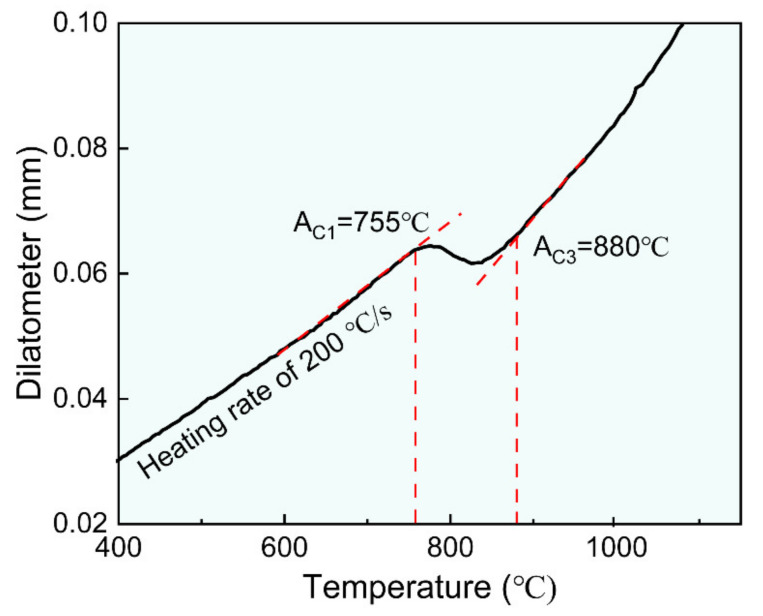
A typical dilatometer curve of the experimental steel.

**Figure 3 materials-16-00550-f003:**
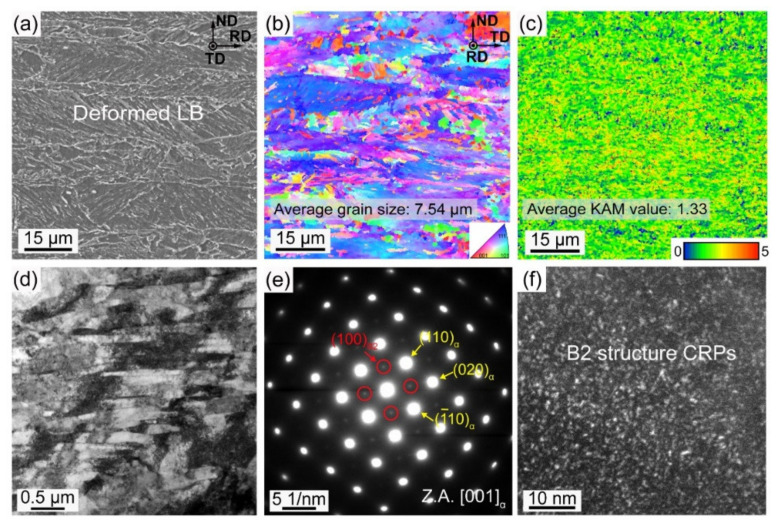
Microstructure features of the experimental steels. (**a**) SEM morphology, (**b**) orientation mapping from EBSD, (**c**) KAM map, (**d**) TEM bright field morphology, (**e**) SAED pattern along [001]_α_, and (**f**) dark field TEM morphology.

**Figure 4 materials-16-00550-f004:**
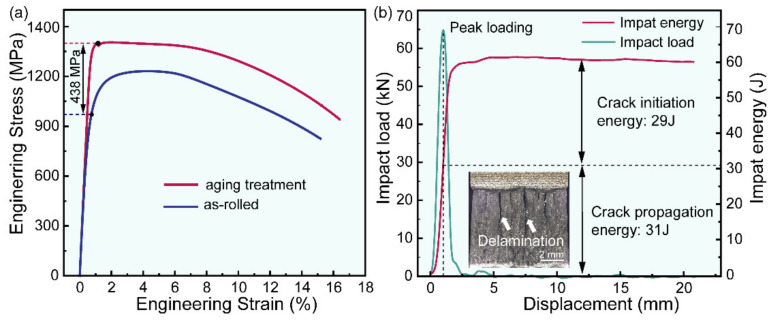
(**a**) Engineering stress–strain curves of experimental steel, (**b**) impact load curve and impact absorbed energy curve of experimental steel. The inset in (**b**) is the stereoscopic observation of the fractured surface.

**Figure 5 materials-16-00550-f005:**
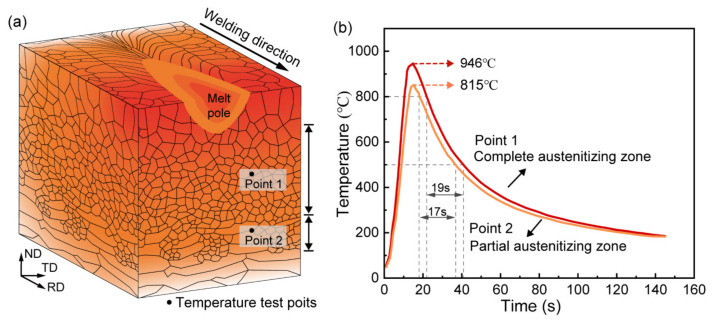
(**a**) Schematic illustration of the welding process and (**b**) thermal cycling curves of different HAZ subregions.

**Figure 6 materials-16-00550-f006:**
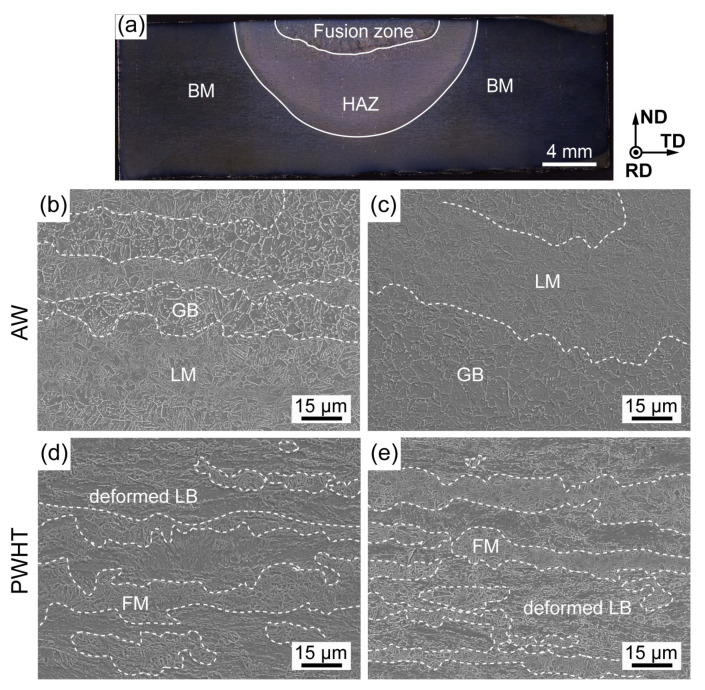
(**a**) The macrograph of cross-section for weld joint, SEM micrographs of (**b**,**c**) complete austenitizing zone and (**d**,**e**) partial austenitizing zone.

**Figure 7 materials-16-00550-f007:**
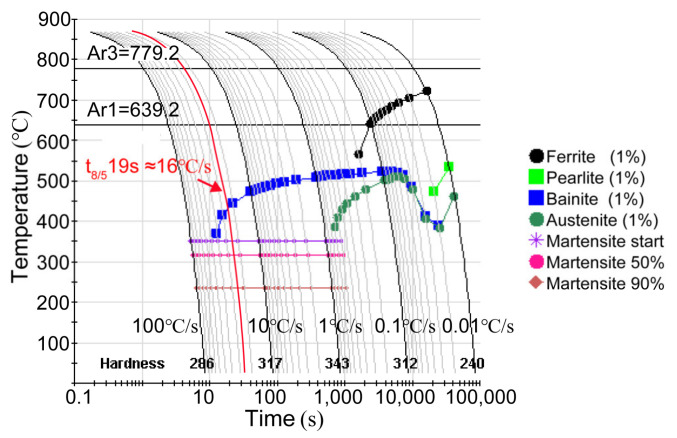
CCT diagram of the investigated steel calculated by the JMatPro database.

**Figure 8 materials-16-00550-f008:**
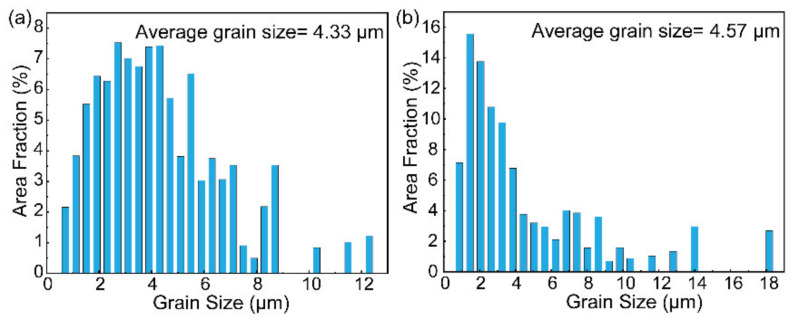
Grain size distribution of the (**a**) complete austenitizing zone and (**b**) partial austenitizing zone.

**Figure 9 materials-16-00550-f009:**
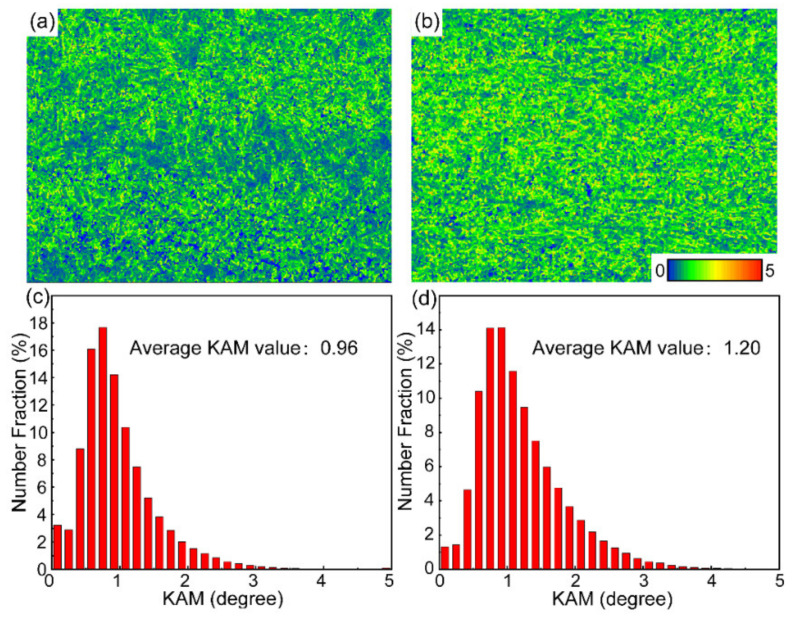
(**a**,**b**) KAM maps, 5° exclusion angle, 1st nearest neighbor, (**c**,**d**) KAM distribution maps. (**a**,**c**) Complete austenitizing zone, and (**b**,**d**) partial austenitizing zone in the AW joint.

**Figure 10 materials-16-00550-f010:**
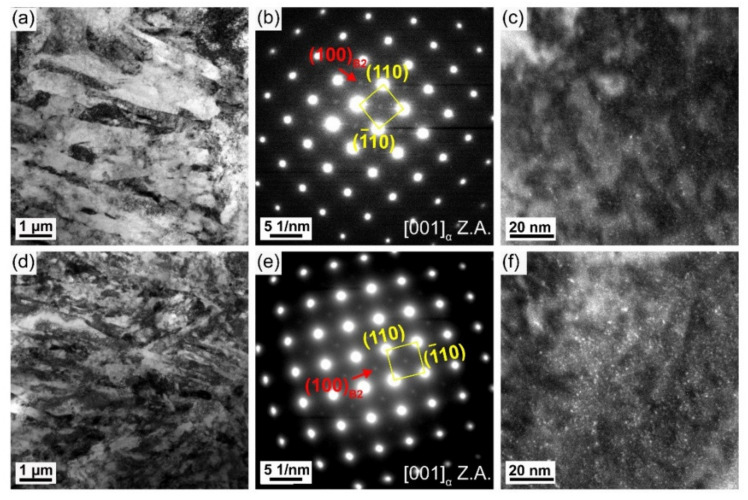
CRP features in (**a**–**c**) the complete austenitizing zone and (**d**,**e**) the partial austenitizing zone of the AW joint. (**a**,**d**) TEM bright field morphology, (**b**,**e**) SAED pattern along to [001]_α_, (**c**,**f**) dark field TEM morphology.

**Figure 11 materials-16-00550-f011:**
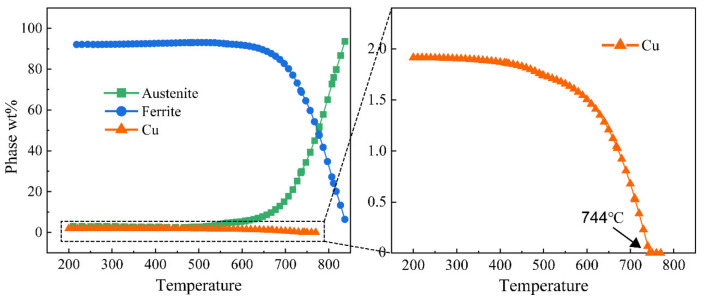
Simulation calculation of phase quality fraction variation with temperature based on the JMatPro database.

**Figure 12 materials-16-00550-f012:**
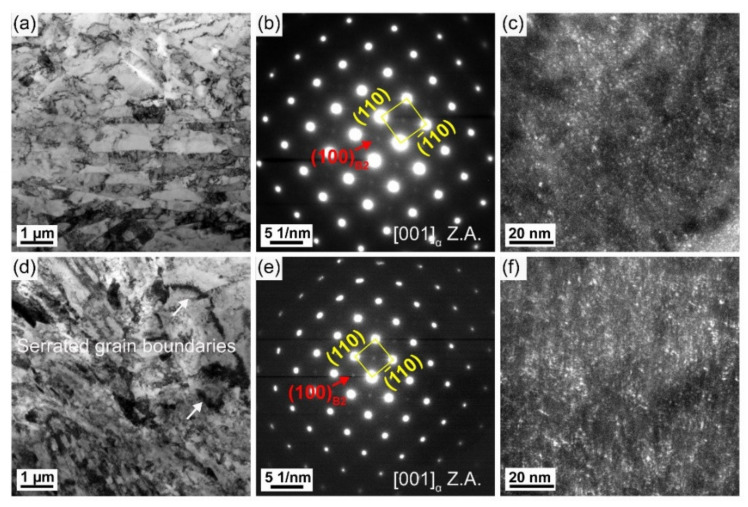
CRP features in the (**a**–**c**) complete austenitizing zone and (**d**,**e**) partial austenitizing zone of the PWHT joint. (**a**,**d**) TEM bright field morphology, (**b**,**e**) SAED pattern along to [001]_α_, (**c**,**f**) dark field TEM morphology.

**Figure 13 materials-16-00550-f013:**
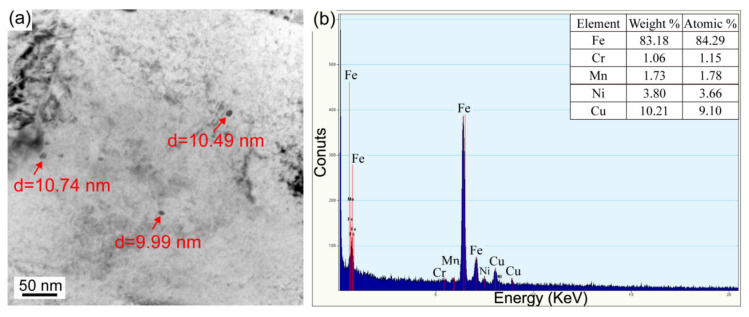
(**a**) TEM bright field morphology and (**b**) EDS result of coarse CRPs (~10 nm) in the partial austenitizing zone of PWHT joint.

**Figure 14 materials-16-00550-f014:**
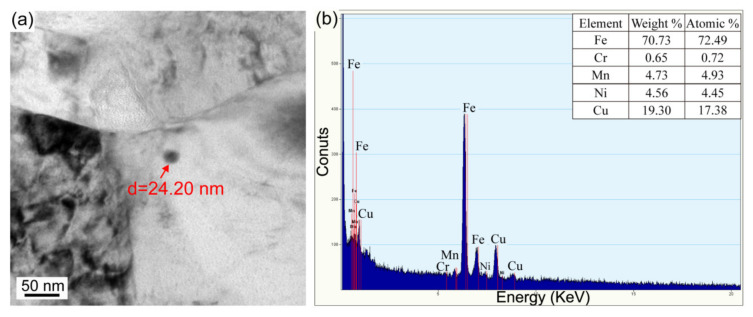
(**a**) TEM bright field morphology and (**b**) EDS result of coarse CRP (~24 nm) in the partial austenitizing zone of PWHT joint.

**Figure 15 materials-16-00550-f015:**
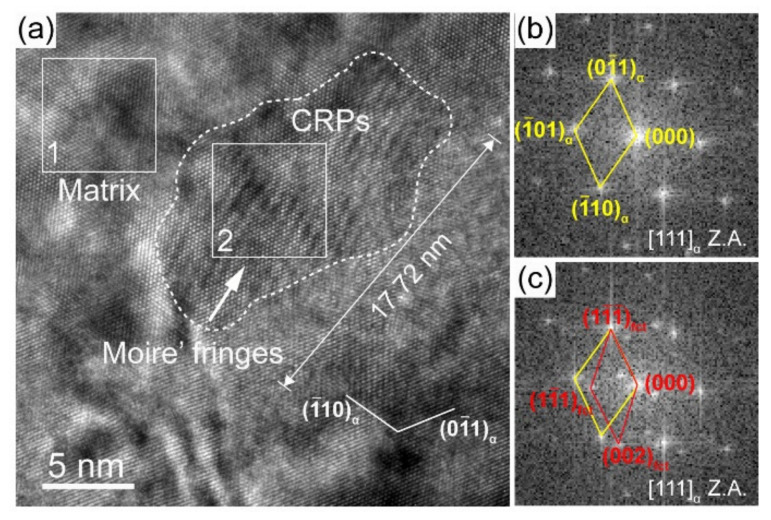
(**a**) High-resolution morphology of coarse CRP (~17 nm) in the partial austenitizing zone of PWHT joint, (**b**,**c**) FFT image of the areas in white square boxes 1 and 2 in (**a**), respectively.

**Figure 16 materials-16-00550-f016:**
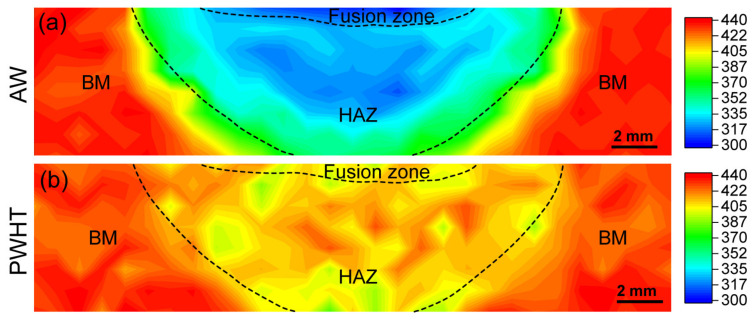
Hardness maps of the (**a**) AW and (**b**) PWHT joints.

**Figure 17 materials-16-00550-f017:**
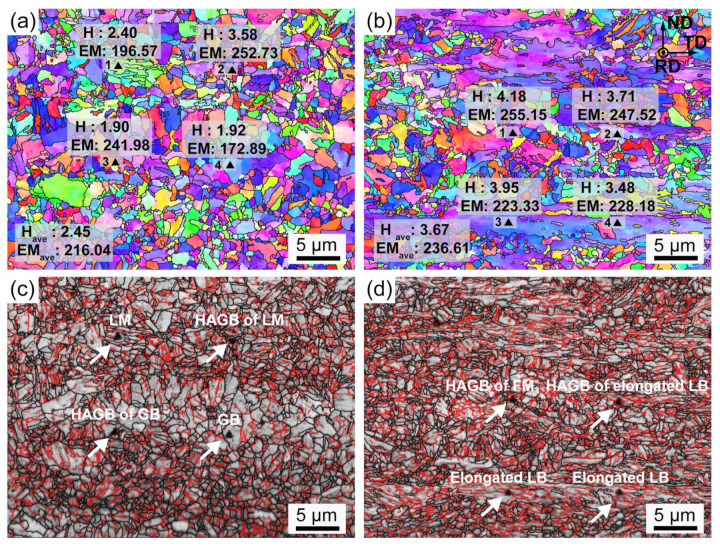
(**a**,**b**) EBSD orientation maps labeled with nano-hardness (H) and elastic modulus (EM) and (**c**,**d**) image quality maps with HAGBs (black line) and LAGBs (red line) for the AW joint. (**a**,**c**) Complete austenitizing zone, (**b**,**d**) partial austenitizing zone.

**Figure 18 materials-16-00550-f018:**
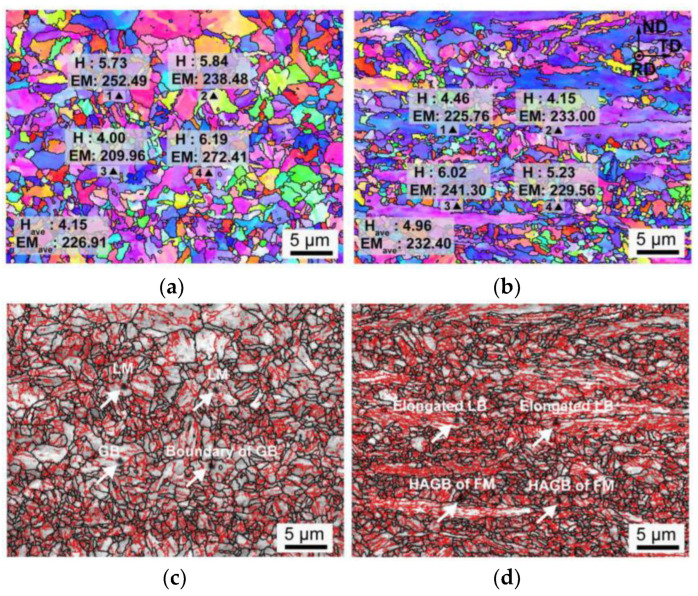
(**a**,**b**) EBSD orientation maps labeled with hardness (H) and elastic modulus (EM) and (**c**,**d**) image quality maps with HAGBs (black line) and LAGBs (red line) for the PWHT joint. (**a**,**c**) Complete austenitizing zone, (**b**,**d**) partial austenitizing zone.

**Figure 19 materials-16-00550-f019:**
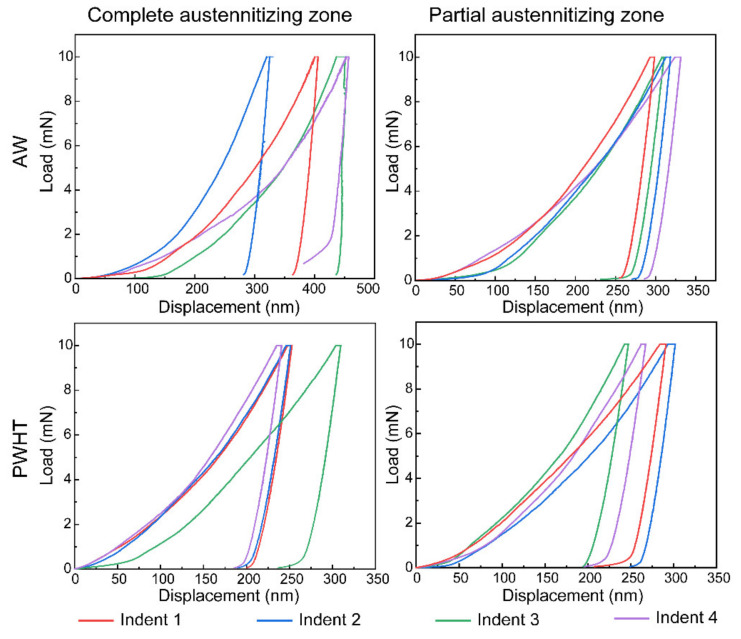
Load–displacement curves of two distinct HAZ regions of different joints.

**Table 1 materials-16-00550-t001:** Chemical composition of the UHS experimental steel (wt%).

C	Si	Mn	Cr	Mo	Ni	Al	Cu	S + P + N	Fe	CE
0.045	0.228	1.05	0.60	0.58	3.77	0.70	1.903	≤0.015	balance	0.295

Carbon equivalent (%) = C + Mn/16 + Ni/50 + Cr/23 + Mo/7 + Nb/5 + V/9.

## Data Availability

Not applicable.

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
