# Peer review of "Characterization of Microstructural Evolution in Heat-Affected Zone of Cu-Bearing Ultra-High-Strength Steel with Lamellar Microstructure"

_materials, 2023, doi:10.3390/ma16020550_

Round 1
Reviewer 1 Report
1. Typos and grammar errors are common in the manuscript. Many sentences are not understandable, some of which can be due to typos or grammar errors. About the manuscript, it is recommended to recheck the grammatical errors or typos.
2.In this manuscript, a discussion of the obtained results must be completely provided. Results should be more discussed and compared to the findings from other researcher.
3.Overall, the results should be reorganized and explained much better.
4.The Conclusion section needs to be rewritten
Reviewer 2 Report
The paper is very interesting and good. Following are my suggestions.
1. Please revise the abstract to include all relevant information.
2. Include extra keywords to assist investigators in finding your research.
3. A table of abbreviations is proposed due to the vast number of abbreviations throughout the article.
4. Include more information about Cu-rich precipitation-strengthened ultra-high strength steel in the introduction.
5. To make your study more appealing to readers and researchers, I recommend adding a new section named "Research significance."
6. Figures require improvement and clarification.
7. Some technical difficulties, such as capitalization inside sentences, commas, superscripts for units, and chemical symbols, must be addressed in the paper.
8. A comparison with past research is required.
9. Consistency in writing hyphenated sentences, whether with or without hyphens, throughout the work. This comment should be applied to all terms that contain hyphens.
10. Include a section on comparisons with past studies.
11. It is ideal to include a section titled "Recommendations" to include the authors' suggestions for future studies.
12. A slight linguistic adjustment is required for the paper.
13. Revise the conclusions in light of past feedback and new adjustments.
14. Cite recent references.
Author Response
Please seen the attachment.

Reviewer 3 Report
Dear Authors,
I have reviewed your paper titled:
"Characterization of microstructural evolution in heat-affected zone of Cu-rich precipitation-strengthened ultra-high strength steel with lamellar microstructure".
The paper fulfils the aims and scope of Materials journal, and can be considered for potential publication. However, I have some suggestions, which are listed below.
General remarks:
- Please add the quantitative results into the abstract.
- I cannot find any proof that you have tested UHS steel. Folowing literature, the border between group of steels is yield point. I cannot find any results of yield point measurements for prepared steel in your paper. The same for tensile strength. Without them, it is hard to assess the paper, because the methodology is characterized by presence of misslacks. Only one curves are presented in Fig. 4, which is not suitable to draw values of mentioned properties. Have you performed more tests? Which values were observed?
- Please use propper way of writting - not "gas-tungsten" but "gas tungsten". The same, should be "ultra-high" not "ultrahigh".
Introduction:
- In my opinion, one important issue is not presented. You should describe problems with welding the UHS steels. Some problems, which have been proved in literature for UHS steels:
- the welding process could produce brittle structures in the HAZ, which leads to decreasing the ductility of the performed joints,
- relatively short fatigue life,
- hydrogen assisted cracking,
- some UHS steels (e.g. S1300) are characterized by high susceptibility to cold cracking.
Materials and Methods:
- Folloging Fig. 1, you have used 50°C/s cooling. Please describe, how this cooling was performed.
- Please use welding engineering terminology in accoance with relevant standards - "arc power"? Should be "heat input" with unit "kJ/mm". Moreover, the UHS steel is very sensitive to the heat input value. Why you have used presented values? It should be presented the same, as full welding parameters - current, voltage and welding speed. Moroever, the gas flow is not presented.
- Some of used tests are standarized. However, I cannot find any information about used standards.
Results and Discussions:
- Please show top view of performed specimens. Moreover, it is worth to show macrographs of crosssection, to show the geometry and proove that none imperfections were observed. Or, you have observed imperfections? If yes, which and where? Even none cracks were observed in HAZ?
-Have you performed any statistical analysis of measured and calculated values?
Conclusions:
- Please support conclusions with the quantitative results.
Round 2
Reviewer 1 Report
The manuscript is accepted in present form.
Reviewer 2 Report
Now, the revised manuscript can proceed for acceptance for publication.
Reviewer 3 Report
Dear Authors,
Paper could be published in this state.
Best regards,